# Electrochemical Biosensors in the Diagnosis of Acute and Chronic Leukemias

**DOI:** 10.3390/cancers15010146

**Published:** 2022-12-26

**Authors:** Alessandro Allegra, Claudia Petrarca, Mario Di Gioacchino, Giuseppe Mirabile, Sebastiano Gangemi

**Affiliations:** 1Division of Hematology, Department of Human Pathology in Adulthood and Childhood “Gaetano Barresi”, University of Messina, 98125 Messina, Italy; 2Department of Medicine and Aging Sciences, G. D’Annunzio University, 66100 Chieti, Italy; 3Center for Advanced Studies and Technology, G. D’Annunzio University, 66100 Chieti, Italy; 4Institute for Clinical Immunotherapy and Advanced Biological Treatments, 65100 Pescara, Italy; 5Unit of Allergy and Clinical Immunology, Department of Clinical and Experimental Medicine, School of Allergy and Clinical Immunology, University of Messina, 98125 Messina, Italy

**Keywords:** electrochemical biosensors, acute leukemia, chronic leukemia, nanoparticle, diagnosis, chemoresistance, biomarker electrochemical biosensors, acute leukemia, chronic leukemia

## Abstract

**Simple Summary:**

Early leukemia diagnosis remains the indispensable goal for effectively treating these diseases. The presence of specific genetic alterations can be helpful for a certain diagnosis. In this regard, biosensors identifying deoxyribonucleic acid molecules as analytes are DNA biosensors. Electrochemical biosensors are a specific form of DNA biosensors that determine the variation in the electrical characteristics of the nano-sensing interface. Furthermore, electrochemical biosensors produced employing different nanomaterials can augment specificity and sensitivity to identify leukemia-related genes, such as BCR/ABL (a fusion gene of chronic myeloid leukemia) or PML/RARα (the promyelocytic leukemia/retinoic acid receptor alpha useful for the diagnosis of promyelocytic leukemia). Thus, the present review reports the preclinical and clinical data existing in the literature on the possible use of such sensors in the diagnosis of leukemic disease.

**Abstract:**

Until now, morphological assessment with an optical or electronic microscope, fluorescence in situ hybridization, DNA sequencing, flow cytometry, polymerase chain reactions, and immunohistochemistry have been employed for leukemia identification. Nevertheless, despite their numerous different vantages, it is difficult to recognize leukemic cells correctly. Recently, the electrochemical evaluation with a nano-sensing interface seems an attractive alternative. Electrochemical biosensors measure the modification in the electrical characteristics of the nano-sensing interface, which is modified by the contact between a biological recognition element and the analyte objective. The implementation of nanosensors is founded not on single nanomaterials but rather on compilating these components efficiently. Biosensors able to identify the molecules of deoxyribonucleic acid are defined as DNA biosensors. Our review aimed to evaluate the literature on the possible use of electrochemical biosensors for identifying hematological neoplasms such as acute promyelocytic leukemia, acute lymphoblastic leukemia, and chronic myeloid leukemia. In particular, we focus our attention on using DNA electrochemical biosensors to evaluate leukemias.

## 1. Introduction

### General Considerations on Electrochemical Biosensors

There are several systems for identifying tumors, but not all methods allow rapid identification of the neoplasm early, allowing an efficacious treatment. The biochemical analyses focused on discovering and evaluating blood biomarkers that are the most suitable for an early diagnosis. However, most of these diagnostic approaches, such as morphological evaluation of the cells, flow cytometry (FCM), fluorescence in situ hybridization (FISH), DNA sequencing, polymerase chain reactions (PCR), and immunohistochemistry, use long-lasting methods with tedious multiple steps. Further limitations of traditional techniques that should not be underestimated are the easy contamination and the use of expensive chemicals. Furthermore, the requirement of a thermal cycler largely limits the applications of PCR in resource-limited settings and point-of-care measurements. On the other hand, electrochemical biosensors have the advantages of small instrument size, simple operation, non-invasive detection, low consumption of reagents and solvents, and high signal-to-noise ratio [1]. To these characteristics, the electrochemical biosensors associate a high sensitivity, which appears particularly useful both in the diagnosis of neoplastic pathologies and in the evaluation of the therapeutic efficacy through the study of the minimal residual disease. 

An electrochemical sensing apparatus might be useful in different areas that comprise environmental checking, food investigation, drug evaluation, and clinical diagnosis [2]. A biological sensor transforms a biological reaction by biological materials into a readable signal [3]. The fundamental rule of a biosensor is that biomolecules are integrated into a matrix that supports the sensing bioanalyte. Elements of the biosensor, such as electrodes and the matrix placed between the detection layer and transducer, have a relevant effect in characterizing the specificity and selectivity of the biosensor [4]. Depending on the type of transducer, the sensors can be distinguished into optical biosensors based on the amount of light or electrochemical biosensors based on electrical distribution. Electrochemical biosensors have broad applicability and are particularly suitable for clinical use. The application of electrochemical biosensors, in fact, not only has greater sensitivity but involves cheaper, non-toxic, and simpler procedures than the optical ones [5].

The electrochemical detection can be amperometric or potentiometric. The first one transforms the analytical data obtained through the bio-recognition procedure into potential. In contrast, the second one evaluates the direct potential current correlated to the oxidation or the decrease in an electroactive species [6].

Furthermore, the specificity and sensitivity of the electroanalytical procedure can be enhanced with the introduction of new electrically conductive substances capable of immobilizing biological constituents on the electrode face, with the ability to detect low-level chemicals in vitro and avoid the possible interference of the various substances present in vivo [7,8,9,10], without a complex sample pre-handling. Furthermore, they can be miniaturized and have shorter execution times than traditional methods. Usually, a three-electrode structure is used in electrochemical target sensing. The working electrode has a central function in the redox system of an electrochemical cell. It can employ several metal electrodes, such as gold, silver, platinum, mercury, glassy carbon, and screen-printed electrodes [11].

Furthermore, when electrodes are constructed with diverse nanomaterials, they can identify several biomolecules with great sensitivity; micro-nanomaterials-funded electrochemical biosensors have displayed high analytical capacity in terms of repeatability, linear range, and limit of detection. Moreover, combining these nanomaterials with biomolecules, such as DNA, can enhance the sensor capability through signal amplification [12]. Significant progress in this regard has been achieved with magnetic and metal nanomaterials [13,14,15,16]. Nanomaterials present great surface area-to-volume proportion, excellent electrocatalytic activity, and augmented adsorption ability [17,18,19]. Concerning traditional biosensors, nanomaterial-adapted electrochemical sensors present good biocompatibility [20,21]. Moreover, crystallographic axis orientation makes nanomaterials particularly suited to their purpose [21]. For instance, graphene, an allotrope of carbon, presents single-atom-thick sheets that are ordered hexagonally sp2-bonded [22]. Furthermore, reduced graphene oxide (rGO) has exceptional electrical conductivity [23] and offers appropriate support for labeling inorganic and organic particles due to great π interactions and hydrophobic properties [24,25,26].

The aim of this review was to analyze published data on the possible use of electronic biosensors for the diagnosis and study of hematological neoplasms and, in particular, the use of DNA electrochemical biosensors in the evaluation of acute and chronic myeloid leukemias.

## 2. Bioreceptor Molecules as Biorecognition Elements

Several bioreceptor molecules, such as antibodies, enzymes, and metal ions, can be used. They are immobilized on the electrodes to augment the signal and improve biomarker identification [27]. The type of bond appears different depending on the structure of the bioreceptor. For instance, antibodies are placed on the electrode plane via covalent bonds such as ester, amide, or thiol [28]. Cells can also be immobilized on the electrode and utilized as a biorecognition layer. In this process, the cell membrane identifies the component of the solution, such as antibodies, aptamers, or small cell vesicles. Several studies on the ability of electrochemical biosensors to identify different forms of cells, such as cancer and bacteria cells, have been carried out [29,30,31].

### DNA Biosensors

The most common molecules used as bioreceptors are nucleotide (DNA or RNA) sequences and single-stranded DNAs. If the corresponding structure is existent in the sample (adenine to thymine and cytosine to guanine), the coupling occurs, and an electrochemical signal is produced [32,33,34,35]. However, this method’s relative disadvantage is the electrolytic suspension’s specificity and stability.

DNA sensors founded on nucleic acid hybridization are presently under evaluation as they are cheap and require unsophisticated equipment [36,37,38]. However, systems can be applied to identify sequence-specific hybridization occurrences directly [39] or by DNA intercalators [40,41,42,43,44], which develop a structure with the nitrogenous bases of DNA. The interface of DNA with small molecules is an essential phenomenon in this case [45]. These substances, which noncovalently interrelate with DNA, have a stable bond with DNA via weak interactions, such as the hydrogen bonding, and van der Waals cohesive properties along the groove of the DNA helix.

Several conducting polymers are physically or covalently modified by nanomaterials, especially nucleic acids, which exhibit catalysis features [46] that can be used in biosensor design [47]. Polymer film-adapted electrodes can generate a significant concentration of functional groups on the face of the electrode and augment the permanence of fixed function groups. This type of immobilized DNA presents unique characteristics of tunable conductivities and easy processing. These render them the better material for ultrasensitive chemical biosensors [48,49] (Figure 1).

Several other approaches have been utilized, favoring the enhancement of the sensitivity of electrochemical identification [50]. For instance, an enzyme-based intensification method has been implemented to enhance the specificity and sensitivity of these biosensors [51].

## 3. Acute Promyelocytic Leukemia

Acute promyelocytic leukemia (APL) is characterized by the presence of a reciprocal chromosome translocation, t(15;17) (q22;q12), causing the fusion between PML and retinoic acid receptor alpha (RARA), which generates PML/RARA fusion gene. APL was classified by the 2016 World Health Organization (WHO) criteria as a distinct entity apart from rare variants of promyelocytic leukemia. This genetic change has a critical role in the onset of APL, and its detection is utilized for the diagnosis [52]. The conventional techniques for the identification of the PML/RARA fusion gene are FISH, chromosome analysis, flow cytometry (FCM), and real-time quantitative reverse transcription polymerase chain reaction (RT-PCR) [53,54,55,56]. However, chromosomal analysis, which helps demonstrate chromosomal changes in configuration and number, is time-consuming and has modest sensitivity. FISH has greater sensitivity but small precision in immobilization. Moreover, even though FCM would detect 5000–10,000 cells, it recognizes the fluorescence characteristics only in a single cellular plane. Finally, RT-PCR identification is sensitive, but false negatives and positives are possible. However, these would be rare with appropriate controls, and the number of chromosome changes is difficult to quantify. Therefore, a novel, efficient technique has been devised to evaluate the presence of the PML/RARA fusion gene. In particular, procedures that use electrochemical sensors have been proposed. They differ in the number and forms of capture probes and the type of nanoparticle used. Each technique has specific advantages and limitations regarding the possibility of application in biological fluids.

Recently, Gamero et al. [57] implemented a sensor employing gold nanoparticles structured with a repetitive square-wave oxidation-reduction cycle (SWORC) as a transducer. Such a nanoporous gold electrode (NPG) transducer was used to identify APL cells. The DNA biosensor employed Methylene Blue (MB) as an electroactive indicator. Differential pulse voltammetry (DPV) was used to check the hybridization reaction on the probe electrode, and the reduction in the peak current of MB was reported upon the hybridization of the probe with target DNA. Results displayed that the peak current was linear with the level of the corresponding strand with an identification limit of 6.7 pM. This novel sensor showed remarkable specificity and sensitivity and has also been applied for a test of PCR real samples with a good result, probably due to the excellent conductivity and the great surface area [58].

However, using nanotechnologies has allowed the implementation of numerous other analysis methods.

Carbon nanotubes (CNTs), having conductivity, electrocatalytic effects, and mechanical strength similar to gold nanoparticles, have been extensively employed in electrochemical biosensor generation [59]. Therefore, a new DNA hybridization biosensor with a CNTs-founded nanostructured membrane was evaluated. The sensing platform was assembled by regularly dispensing FePt nanoparticles (NPs). By coupling the benefits of FePt NPs and CNTs, this structure significantly accelerates the electron-transfer procedure and has an exceptional sensitivity for DNA identification (detection value of 2.1 1013 mol/L). This DNA electrochemical biosensor was extremely useful in distinguishing single-base or double-base mismatched sequences [5]. Furthermore, this procedure is straightforward as it does not necessitate advanced labeling of oligonucleotide probes [60].

Another possible nanomaterial that could be employed to modify electrodes is GO. It possesses several oxygen functional groups offering connecting places for the covalent immobilization of DNA. Metal nanoparticles, quantum dots (QDs), and metal oxide nanomaterials have been employed to functionalize GO. These modified its sheet resistance, reduced electronic conductivity, and augmented the electrochemical functions. QDs are recognized as low-priced and easy-to-prepare [61], with exceptional charge transport mobility and electrical conductivity [62,63,64,65,66,67,68].

An electrochemical sensor based on QDs-modified graphene oxide (QDs/GO) nanomaterials with glassy carbon electrodes (QDs/GO/GCE) was studied for PML/RARα fusion gene identification [69]. It showed better electrochemical function and more significant affinity to the DNA sequences among the original QDs or GO, with an extraordinary sensing performance (identification limit of 83 pM) [69].

Wei et al. [70] assembled a poly-calcon carboxylic acid (poly-CCA) film-adapted electrode by cyclic voltammetry (CV) to prepare an electrochemical DNA sensor for the identification of the PML/RARA fusion gene. It employed an 18-mer single-stranded deoxyribonucleic acid as the capture probe. This probe was covalently bound via free amines on the DNA bases. Furthermore, in this case, the results demonstrated that the oxidation peak current was linear with the number of complementary strands (identification limit of 6.7 × 10^−13^ M). This technique could be applied in clinical practice [70].

Using natural substances to increase the technique’s sensitivity is also interesting. Aloe-emodin (1,8-dihydroxy-3-[hydroxymethyl]-anthraquinone) is an active substance extracted from the *Rheum palmatum* L. (Polygonaceae) [71]. The analysis of the Aloe-emodin (AE) relationship with salmon sperm DNA at the DNA-modified GCE demonstrated that AE had an extraordinary electrochemical effect on the GCE. AE could intercalate into DNA strands generating a non-electroactive structure that causes the diminution of the reduction peak current of AE. The variance between AE at ss- and dsDNA has been utilized to generate a sequence-restricted DNA sensor for discovering the PML-RAR fusion gene (identification limit of 6.7 × 10^−8^ M), which could have a possible application in APL diagnosis [72].

However, an exciting feature in the investigation of electrochemical sensors is the opportunity to exploit molecules different from DNA as a target. For instance, locked nucleic acid (LNA) nucleotides include a methylene covalent bridge between the 4-carbon and the 2-oxygen of the ribose portion. This bridge efficiently ‘locks’ the ribose in the N-type configuration that is prevalent in A-form DNA and RNA. For this reason, LNAs differ from DNA as the covalent bridge provokes a greater affinity for complementary DNA and RNA sequences. Furthermore, LNA/DNA hybrids are more destabilized by single-base mismatches than comparable DNA/DNA hybrids. Finally, LNA nucleotides display minor toxicity and augmented triplex generation [73,74,75].

According to these characteristics, an LNA changed probe was used to detect target DNA and establish that LNA probes hybridized with affinity to complementary targets; data also showed an excellent specificity to distinguish the targets that diverge by a single-base mismatch.

Thus, a novel biosensor was planned that utilizes a hairpin LNA probe double-marked with carboxyfluorescein molecule (FAM) and biotin [76]. The probe was immobilized at a streptavidin-adapted electrode surface. Target joining opened the hairpin of the probe, which presented a structural change pushing FAM away from the electrode. In this way, the FAM label turned out to be reachable by the anti-FAM-HRP, and the target hybridization phenomenon could be transduced through the enzymatically intensified electrochemical current signal. This novel sensor revealed its brilliant specificity and sensitivity (detection limit 83 fM), targeting DNA even when proven to employ biological fluids [76]. Furthermore, the DNA biosensor could successfully keep away non-specific adhesion of proteins, becoming a perfect biosensor for clinical appliances.

Finally, some studies attempted to find new methodological solutions to render the method more precise. The DNA originating from human blood generally is native double-stranded (ds-) fragments, which can be more stable in vitro. This could hinder the hybridization between probes and targeting DNA. For this reason, diverse pre-treatment techniques have been employed to denature the dsDNA fragments and attain ss-DNA target fragments. However, these procedures might hamper the detection during successive probe-target hybridization. Thus, experimentation has designed a different sensing approach founded on dual-probe identification, employing two DNA probes corresponding to the two constituents from target dsDNA. These two DNA probes adapted onto the two different electrodes could identify the corresponding sequences of dsDNA in the same procedure after thermal treatment. Thus, a sensor planned on a dual-probe organization can efficiently decrease the effect of reannealing the two diverse components of dsDNA and enhance the efficacy of probe–target hybridization.

A type of dual-probe E-DNA (DE-DNA) sensor has been employed to identify the dsDNA of the PML/RARα gene based on the “Y” junction structure [77]. Nevertheless, even though it has been demonstrated the high sensitivity of the sensor with LNA probes [77], its wide clinical use is limited by the high cost and complex techniques.

A better specificity seems to be guaranteed by employing a diverse type of probe. In a study, a new dual-probe, containing two groups of 2′-fluoro ribonucleic acid (2′-F RNA), was prepared to detect dsDNA of the PML/RARα gene [78]. The two sets of 2′-F RNA adapted probes complemented the corresponding strand from target dsDNA. In addition, the biotin-adapted enzyme, which causes the electrochemical current signal, was placed in the plane via an affinity connection between streptavidin and biotin. This biosensor had a detection limit of 84 fM target dsDNA and displayed good specificity. In fact, this sensor’s specificity was evaluated in an assorted hybridization suspension comprising diverse types of mismatch sequences. The technique showed exceptional specificity in these conditions [78] (Table 1).

## 4. Acute Lymphoblastic Leukemia

An important series of studies have been conducted on electrochemical biosensors for diagnosing acute lymphoblastic leukemia (ALL).

ALL, due to the clonal growth of B and T progenitors in the bone marrow (BM), in the blood, and extramedullary places, is the most frequent tumor in children. Once believed to be an incurable condition, the 5-year survival percentage of pediatric ALL has improved significantly over the past years. Now, it is beyond 90% [79].

In this field, biosensors have employed several targets to recognize neoplastic cells or genetic alterations such as Pax-5. An aptamer-founded electrochemical sensor has been constructed [80], immobilizing the thiolated sgc8c aptamer on gold nanoparticles-coated magnetic Fe_3_O_4_ nanomolecules and used to detect tumoral cells. After the presentation of the ALL cells on the Apt-GMNPs, the hairpin configuration of the aptamer is modified and the intercalator molecules (ethidium bromide) are liberated. These cause a reduction in the electrochemical signal. Furthermore, the use of nitrogen-doped graphene nanosheets on the electrode plane amplifies the read-out signal displaying a linear response over leukemia cells from 10 to 1 × 10^6^ cell mL^−1^. This biosensor was effectively employed for the identification of leukemia cells also in complex media such as human plasma.

A study confirmed that the nitrogen-doped graphene displayed a better result than the original graphene, and the utilization of gold nanoparticles substantially increased the possibility of identification (detection limit 10 cells mL^−1^) differentiating ALL cells and control (Ramos) cells. This approach is a promising strategy and could be employed to identify other types of cancer cells [80].

A further possible target for these biosensors is the Pax-5 gene. This is the only paired box (PAX) family component in the hematopoietic system, and it can be classified into Pax-5a, Pax-5b, Pax-5c, Pax-5d, and Pax-5e. Pax-5a is the most relevant, being essential for B-cell growth and differentiation [81], while an altered expression can provoke B lymphocytic leukemia [82,83]. A new electrochemical sensor to identify Pax-5 gene mutation is based on G-quadruplex. The G-quadruplex is a particular form of DNA secondary structure. In this case, the four guanine bases are combined end-to-end to generate a G-tetrad, and the contiguous G-tetrads generate a G-quadruplex by π-π stacking [84,85]. By connecting to different little particles, G-quadruplex DNAzyme presents great stability. For this reason, G-quadruplex DNAzyme is broadly utilized in electrochemical sensors [86]. For instance, the G-quadruplex/hemin complex produced by a G-quadruplex connecting to hemin presents a peroxidase catalytic function and can be used as an electrochemical sensor [87]. Therefore, an electrochemical procedure was constructed to identify the ALL Pax-5a gene employing an enzyme-supported signal intensification to produce a G-quadruplex/hemin DNAzyme [88].

Because the restriction enzymes Nt.BbvCI and Klenow fragment polymerase, the presence of Pax-5a modifies the hairpin structure, while a more significant number of G-quadruplex sequences are generated. These sequences form a G-quadruplex/hemin complex on the plane of the electrode, and the electrochemical identification of Pax-5a is obtained through the G-quadruplex/hemin complex-catalyzed reduction of H_2_O_2_ (identification limit 4.6 fM). Moreover, it is remarkable that this enzymatic reaction happens out of the electrode, which increases the effectiveness of the response and simplifies the procedure. In fact, the interference of proteins and nucleic acid strands on the electrode surface is avoided.

### Electrochemical Biosensors and Chemoresistance

In the near future, the electrochemical biosensors will be applied not only for the diagnosis of hematological malignancies but also for the identification of chemoresistance. A new electrochemical biosensor was designed to rapidly identify a drug-resistant leukemia cell line (K562/ADM cells—Adriamycin resistant derivative of K562 cell line) obtained by lymphoblasts originating from a chronic myeloid leukemia patient.

This biosensor is based on the P-glycoprotein (P-gp) presence on the cell membrane [89]. An interface of gold nanoparticles/polyaniline nanofibers (AuNPs/PANI-NF) was used to prepare the sensor. Au/PANI-NF-based sensors covered with anti-P-glycoprotein (anti-P-gp) molecules could detect the presence of P-gp cells. Due to the great affinity of anti-P-gp for K562/ADM cells, this sensor displayed an extraordinary analytical ability with an identification limit of eighty cells per milliliter. A flow cytometric technique was used to evaluate the efficacy of the biosensor, demonstrating its usefulness for clinical purposes [89].

## 5. Chronic Myeloid Leukemia

Biosensors have been tested not only in acute but also in chronic myeloid leukemia (CML). CML is a myeloproliferative tumor of stem cells that represents 14% of all cases of leukemia in the United States. More than 95% of CML patients are characterized by a reciprocal translocation involving chromosomes 9 and 22. This causes the generation of an irregularly short chromosome 22, called the Philadelphia chromosome (Ph). This chromosome leads to the BCR-ABL1 gene fusion, which provokes an abnormal growth of leukemic cells [90].

Among different nanoparticles, cerium dioxide (CeO_2_) has been rated useful for biosensor generation and to immobilize biotargets due to its chemical stability, biocompatibility, and great adsorption capacity [91,92]. In particular, CeO_2_ might be advantageous for immobilizing negatively charged molecules due to its high isoelectric point. CeO_2_ nanoparticles are employed as immobilizing carriers of DNA probes, and an effective DNA electrochemical sensor was designed based on GoldNPs generated at the surface of carbon nanotubes (MWCNTs), CeO_2_, and chitosan (Chits) membrane to detect BCR/ABL fusion genes in CML [93]. Due to the synergistic action of CeO_2_ particles with a relevant adsorption capacity and MWCNTs with an important electron relocation capacity, this sensor demonstrated a relevant sensitivity (range from 1 × 10^−9^ M to 1 × 10^−12^ M; identification limit of 5 × 10^−13^ M [93]. This technique was also successfully utilized for identifying PRC real samples with relevant results and displayed excellent selectivity and a relevant stability.

In a different study, a hybrid nanocomplex composed of chitosan and zinc oxide nanomaterials (Chit-ZnONP) immobilized on a polypyrrole (PPy) layer was used [94]. DNA sections were covalently restrained, permitting molecular identification. The sensor performance was evaluated employing recombinant plasmids inclosing the specific oncogene and biological samples from CML subjects. This sensing system showed great specificity and selectivity (detection limit 1.34 fM) [94]. Moreover, the analysis is simple, rapid and useful for early tumor diagnosis; this nanostructured system might be a valid option for the genetic detection of the BCR/ABL fusion gene. Modest amounts of oligonucleotides and biological samples are necessary for the analysis, so this system can be easily used in the clinical setting.

MXene is a novel substance evaluated to make sensors detect CML cells. MXene is a bi-dimensional material that presents good biocompatibility, appropriate electrical conductivity, and mechanical strength, which render MXene a perfect structure for biosensing for electrochemical use [95,96]. MXene-based materials have been employed as electrocatalysts for identifying drugs, pollutants, and biomarkers [97].

The electrocatalytic activity of Ti_3_C_2_T_X_ MXene for phenols oxidation was useful in creating an efficient catalytic amplification system for the electrochemical sensor to identify the BCR/ABL fusion gene [98]. Ti_3_C_2_T_X_ MXene was decorated with gold nanoparticles for DNA capture. Further, a DNA walking machine was used to identify BCR/ABL fusion genes and facilitate nucleic acid amplification. This biosensor achieved a great sensitivity detection limit down to 0.05 fM. Different amounts of BCR/ABL fusion gene were combined with ten-fold serum samples and investigated with this biosensor to evaluate the applicability of this sensor in clinical practice. Adequate recovery rates were achieved, oscillating from 93.60% to 110.42%. Finally, a direct comparison between the results obtained with this biosensor and with RT-PCR demonstrated the technique’s validity [98], showing its possible application in clinical settings.

However, the specificity and sensitivity of biosensors differ depending on the diffusion coefficient and the electroactive surface area. The use of QDs can improve several characteristics of biosensors. For example, a QDs-modified electrode has been employed as a transducer plane for covalent immobilization of a probe oligonucleotide to evidence the BCR-ABL fusion gene [99]. This form of biosensor demonstrated a substantial improvement in its mismatch detection ability with respect to a biosensor planned without QDs (detection limit 1.0 pM), suggesting encouraging possibilities for clinical investigations [99] (Table 2).

### Monitoring of the Efficacy of the Therapy through Biosensors

Reduction in the BCR-ABL1 function through the administration of specific tyrosine kinase inhibitors (TKIs) such as imatinib has drastically changed the therapy of CML. Dasatinib, a different TKI, can block the activities of the ATP-binding site of BCR/ABL, provoking programmed cell death of CML cells and inhibiting tumor cell growth [100]. However, the use of dasatinib is still hindered by the capacity of CML cells to acquire drug resistance, which can be prevented by combining Dasatinib with TNF-related apoptosis-inducing ligand (TRAIL), a component of the TNF superfamily. This element has been reported to kill neoplastic cells by connecting to their death receptors, Death Receptor 4 and Death Receptor 5. These receptors promote the onset of programmed cell death in different neoplastic cell lines [101]. Furthermore, TRAIL evidenced synergism with numerous chemotherapeutics [102]. Therefore, an electrochemical cytosensing method to evaluate the efficacy of dasatinib and TRAIL, by identifying the presence of caspase-3 in apoptotic CML cells, was studied [103]. The biosensor was constructed employing a GCE adapted with AuNPs, poly (dimethyl diallyl ammonium chloride) (PDDA), and CNTs [103]. The findings suggested that both Dasatinib and TRAIL stimulate the programmed cell death of CML cells, but their combined use produced better therapeutic results, representing a new possibility for CML treatment. Furthermore, this specific electrochemical biosensor may be easily employed to monitor other tumor therapeutic results thanks to its simplicity, high reproducibility, and great stability.

## 6. Future Perspectives

Several studies have converged on microRNAs (miRNAs) as a relevant tumor marker in recent years. MiRNAs are present in large quantities in biological liquids and are easily identified. Furthermore, the presence of miRNAs is also related to the existence and progress of different diseases, such as diabetes and cardiological diseases, other than tumors [104]. The traditional techniques employed to identify and quantify miRNAs are Northern blot methods, RT-qPCR, and DNA microarray. Generally, these techniques present a high specificity, but the procedures are complicated, particularly expensive, remarkably long, and necessitate experienced personnel.

New strategies based on multifunctional nanomaterials and oligonucleotides for miRNA detection were described [105,106,107]. In particular, electroactive species-labeled DNA probe sequences were employed for miRNA identification. These species can be nanohybrids of inorganic substances, such as cadmium, gold or silver nanoparticles, and organic substances such as ferrocene, thionine, and methylene blue, used as RedOx probes [108,109,110,111,112,113,114,115].

Some screening experiments in vitro have been made to identify microRNAs specific to lymphoblastic leukemia that enabled the discovery of miRNA-128 as a biomarker. Therefore, an electrochemical nanocomposite aptasensor was constructed to detect miRNA-128. To immobilize the aptamer chains on the plane of the GCE, gold nanoparticles/magnetite/reduced graphene oxide (AuNPs/Fe_3_O_4_/RGO) were employed [116]. This electrode was shown to have remarkable selectivity for miRNA-128 among other miRNA molecules [116].

In the future, electrochemical biosensors may be extended and used in several hematological pathologies (Table 3).

An extremely specific biosensor for the non-Hodgkin lymphoma gene was made using electrochemical impedance spectroscopy [117]. The biosensor was based on electrospun NBR rubber, inserted with poly(3,4-ethylenedioxythiophene). The biosensor demonstrated an exceptional detection limit of 1 aM (1 × 10^−18^ mol/L) for the non-Hodgkin lymphoma gene, more than 400 folds smaller than a thin-film equivalent. The sensor displayed extraordinary selectivity, with only 1%, 2.7%, and 4.6% of the indicator recorded for the fully non-complementary sequences, or T–A and G–C base mismatches, respectively [117]. Highly improved selectivity is suggested to be due to negatively charged carboxylic acid moieties (from PAA grafts) that are proposed to electrostatically repulse the non-complementary and mismatched DNA sequences, overwhelming the non-specific attachment.

Moreover, electrochemical sensors could have a strategic role in the clinical use of this analysis, so-called liquid biopsy. For example, evaluating circulating free DNA (cfDNA) transported by blood might be an attractive option to traditional bone marrow biopsy for assessing neoplastic load and offering a complete report of the temporal and spatial heterogeneousness of the tumor genetic landscape [118]. Concerning other cancer biomarkers, cfDNA has several advantages, such as high specificity and little invasiveness [119]. However, the currently available identification methods, such as PCR and DNA sequencing, require difficult, expensive, and long procedures, which limit their clinical application. Electrochemical biosensors are sensitive, specific, and more accessible and are quicker methods of identifying cfDNA [119]. Furthermore, selecting bioreceptors using intensification methods with nanoparticles and multi-tags components might augment the efficacy of electrochemical biosensors.

## 7. Conclusions

At present, electrochemical detection is facing several problems that should be solved to develop a specific and sensitive method for the identification of leukemia. A recent work by Li compared the sensitivity ranges of various techniques for assessing minimal residual disease of hematologic neoplasms [120]. Flow cytometry showed a sensitivity ranging between 10^−3^ and 10^−5^, quite similar to digital PCR, while RT-qPCR for gene fusions presented a sensitivity ranging between 10^−4^ and 10^−5^. For instance, RT-PCR detected the PML-RAR alpha transcript in samples containing up to 0.01 ng of total leukemic RNA, while in CML patients, digital PCR was able to identify one cell in up to 100,000 cells. Hence, by comparing these data with the detection limits reported in Table 1 and Table 2, electrochemical biosensors reported in Table 1 and Table 2 allow the identification of hematological neoplasms with a higher order of sensitivity. The stability of the biosensor is an essential factor for single-use electrodes and for those which are to be utilized repetitively. A different and more critical question for the future improvement of electrochemical biosensors is the appropriate in vivo evaluation of real samples. These difficulties have so far prevented the extensive clinical use of these techniques, which are still mainly used in the experimental ambit. In general, the optimal in vivo sensor should be safe, compatible with cells and tissues, stable, and sensitive. An electrochemical biosensor should have excellent specificity and should use have component detection of several small molecules such as folic acid or aptamers. Furthermore, the combined use of the specific identification capacities of aptamers and the catalytic capacity of enzymes is a potent tool in bioresearch to attain an extremely sensitive and discerning identification of different molecules. Enzymes connected to biomolecules ease the greatly amplified discovery of electroactive compounds [121]. However, the biocompatibility of these biosensors can be maximized by using nanomaterials that are non-toxic, with low reactivity to proteins, that do not stimulate an immune response.

In conclusion, experimental studies show that electrochemical sensors seem very attractive concerning reported techniques such as PCR and FISH, as biosensors or genosensors are very cost-effective compared to these methods [122].

In the future, we will see the pre-eminence of electrochemical biosensors with respect to other methodologies for the identification of leukemia if the in vivo studies will confirm the experimental results and further increase the performance of nanoparticle-based electrodes.

In the future, nanoparticle-based electrochemical biosensors might become a pre-eminent methodology for identifying leukemia if they are proven to be highly sensitive and well-performing detection tools also in in vivo studies.

## Figures and Tables

**Figure 1 cancers-15-00146-f001:**
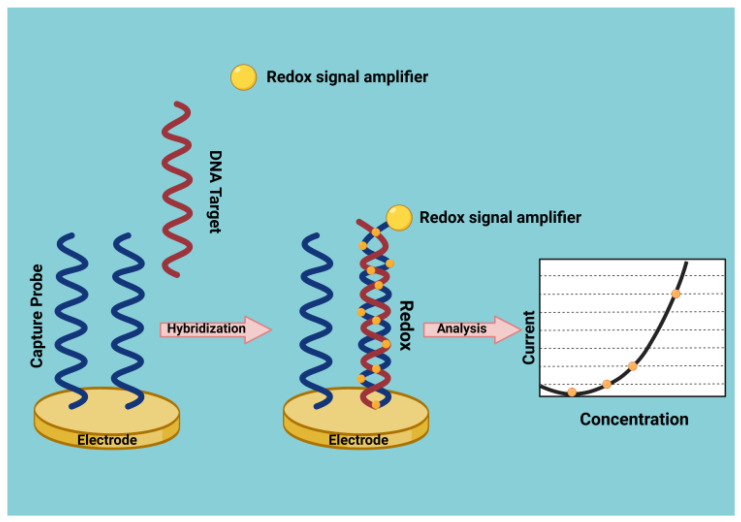
Method for the electrochemical detection of DNA hybridization.

**Table 1 cancers-15-00146-t001:** Possible biosensors for PML/RARα detection.

Disease	Sensor	Detection Limit	Ref.f.
Acute promyelocytic leukemia	Gold nanoparticles	6.7 pM	[58]
target DNA (22-base sequence S2)-5′-CGG GGA GGC AGC CAT TGA GAC C-3′			
immobilized probe (22-base sequence S1)-5′-SH-GGT CTC AAT GGC TGC CTC CCC G-3′			
	Carbon nanotubes	2.1 mol/L	[60]
probe DNA (ssDNA): 5′-TCT CAA TGG CTG CCT CCC-3′;			
target DNA (cDNA): 5′-GGG AGG CAG CCA TTG AGA-3′;			
	Quantum dots/Graphene oxide	83 pM	[69]
capture probe DNA (22-base sequence): 5′-NH_2_-GGTCTCAATGGCTGCCTCCCCG-3′			
complementary target DNA (22-base sequence): 5′-CGGGGAGGCAGCCATTGAGACC-3′			
	Poly-calcon carboxylic acid	6.7 × 10^−13^ M	[70]
immobilized probe (18-base sequence, S1)-5′-NH2-TCT CAA TGG CTG CCT CCC-3′			
target (S2)-5′-GGG AGG CAG CCA TTG AGA-3′;			
	Aloe-amodin/Glassy carbon electrode	6.7 × 10^−8^ M	[72]
immobilized probe(18-base sequence S1)-5′-NH3 TCT CAA TGG CTG CCT CCC-3′			
target (18-base sequence S2)-5′-GGG AGG CAG CCA TTG AGA-3′			
	Gold nanoparticles	84 fM	[78]

**Table 2 cancers-15-00146-t002:** Possible biosensors for BCR/ABL detection.

Disease	Sensor	Detection Limit	Ref.
Chronic Myeloid leukemia	Cerium dioxide, Carbon nanotubes, Chitosan	5 × 10^−13^ M	[93]
Immobilized probe HS-ssDNA (S_1_): 5′-HS-AGA GTT CAA AAG CCC TTC-3′			
Target ssDNA (S_2_, complementary to S_1_): 5′-GAA GGG CTT TTG AAC TCT-3′			
	Chitosan, Zinc oxide nanoparticles	1.84 fM	[94]
Recombinant plasmid containing the BCR/ABL fusion gene: 5′-AGCTTCTCCCTGACATCCGTG-3′			
	MXene, Gold nanoparticles	0.05 fM	[98]
Capture probe SH-TTTCCGGAGGAGCTACCTACGATCAATCCA			
Detection probe ACCACACGCTCCTCCGGCTTT-Biotin			
	Quantum dots	1.0 pM	[99]

**Table 3 cancers-15-00146-t003:** Some of the patented (nano)biosensors that could be used in the diagnosis of hematological diseases.

Patent Publication n.	Publication Date	Title
CN101928767A	2010-12-29	Electrochemical DNA biosensor for detecting BCR/ABL fusion gene of chronic myeloid leukemia (CML)
CN101705279A	2010-05-12	Nano biosensor for detecting PML/RAR alpha fusion gene of acute promyelocytic leukemia
US5871918A	1996-06-20–1999-02-16	The University Of North Carolina At Chapel Hill Electrochemical detection of nucleic acid hybridization
CN102676638A	2011-03-08–2012-09-19	Method and kit for detecting drug-resistance mutation site of ABL kinase domain of BCR/ABL fusion gene
WO2017114008A1	2015-12-30–2017-07-06	BCR gene and ABL gene detection probe, preparation method, and reagent kit
CN103063715A	2012-11-03–2013-04-24	Method for detecting surviving gene based on graphene-gold composite material electrochemical DNA (Deoxyribose Nucleic Acid) biosensor
CN101928767A	2010-12-29	Electrochemical DNA biosensor for detecting BCR/ABL fusion gene of chronic myeloid leukemia (CML)
CN101705279A	2010-05-12	Nano biosensor for detecting PML/RAR alpha fusion gene of acute promyelocytic leukemia
CN2009101120902A	2009-06-29	Electrochemical DNA biosensor for detecting BCR/ABL fusion gene of chronic myeloid leukemia (CML)
US20210048405A1	2021-02-18	Binding probe circuits for molecular sensors

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
