# Peer review of "Electrochemical Biosensors in the Diagnosis of Acute and Chronic Leukemias"

_cancers, 2022, doi:10.3390/cancers15010146_

Round 1

Reviewer 1 Report

The paper by Allegra et al. it is very interesting because it describes in detail a new technology for the diagnosis and monitoring of acute and chronic hematological malignancies. The authors concisely describe the technological and scientific aspects of the biosensor methodology and subsequently describe the applications of this technology in some types of leukaemia.

From a methodological point of view, the authors should explain the advantages of this technology in terms of sensitivity compared to traditional PCR or flow cytometry techniques in the different models used: APL, CML, and ALL. It would be useful to report a table with the different diagnostic sensitivities of the methods used.

Author Response

Reviewer 1

The paper by Allegra et al. it is very interesting because it describes in detail a new technology for the diagnosis and monitoring of acute and chronic hematological malignancies. The authors concisely describe the technological and scientific aspects of the biosensor methodology and subsequently describe the applications of this technology in some types of leukaemia.

From a methodological point of view, the authors should explain the advantages of this technology in terms of sensitivity compared to traditional PCR or flow cytometry techniques in the different models used: APL, CML, and ALL. It would be useful to report a table with the different diagnostic sensitivities of the methods used.

In the conclusions we have dedicated a large paragraph to the comparison of the sensitivity of the various techniques.

Reviewer 2 Report

Allegra et al have systematically reviewed the diagnosis of leukemias using various forms of Electrochemical biosensors in the article entitled “Electrochemical biosensors in the evaluation of acute and chronic leukemias”. Authors have also highlighted the application of biomolecules as biorecognition elements in Electrochemical biosensors for cancer detection. This review is interesting, but the article has many grammatical and sentence errors. The language organization needs to be improved. In addition, the authors should do the following before acceptance.

1.      Title looks very general. Authors may modify the title with important keywords like detection, or leukemia diagnosis.

2.       Authors need to improve the references by citing recent references like

https://doi.org/10.1016/j.ab.2022.114736

https://doi.org/10.1016/j.bioelechem.2022.108176

https://doi.org/10.1016/j.electacta.2021.138863

https://doi.org/10.1016/j.bioelechem.2022.108176

3.      Table may be provided for DNA sequences used as biorecognition elements in the DNA biosensors

4.      Authors may provide Information about the commercialized or patented Electrochemical biosensors for the detection of various cancers

5.      Typographical errors can be avoided. The language and grammar used throughout the manuscript need to be improved

Author Response

Reviewer 2

Allegra et al have systematically reviewed the diagnosis of leukemias using various forms of Electrochemical biosensors in the article entitled “Electrochemical biosensors in the evaluation of acute and chronic leukemias”. Authors have also highlighted the application of biomolecules as biorecognition elements in Electrochemical biosensors for cancer detection. This review is interesting, but the article has many grammatical and sentence errors. The language organization needs to be improved. In addition, the authors should do the following before acceptance.

  1. Title looks very general. Authors may modify the title with important keywords like detection, or leukemia diagnosis.

We modified the title:

Electrochemical biosensors in the evaluation diagnosis of acute and chronic leukemias.

  1. 2.      Authors need to improve the references by citing recent references like

https://doi.org/10.1016/j.ab.2022.114736

https://doi.org/10.1016/j.electacta.2021.138863

https://doi.org/10.1016/j.bioelechem.2022.108176

We have added two of the required bibliography as one (https://doi.org/10.1016/j.bioelechem.2022.108176) was already entered.

  1. Table may be provided for DNA sequences used as biorecognition elements in the DNA biosensors

We have added the required data in the tables when available

We have added a table with the names of some of the biosensors that could be used in the diagnosis of hematological diseases

EP20 EP2017612A1

EP2017612A1

  1. Typographical errors can be avoided. The language and grammar used throughout the manuscript need to be improved

We subjected the work to linguistic editing

Reviewer 3 Report

The manuscript by Allegra et al. is a review of electrochemical biosensors for leukemia. There is also a focus on nanomaterials being used in these biosensors. Overall, the ideas and concepts are important and worthy of review. However, this manuscript is poorly written and difficult to follow. The following are more specific critiques.

1. The manuscript is very poorly written, which makes it difficult to understand. This is the case throughout the manuscript, but is worse in the abstract, introduction, and conclusions sections. The entire manuscript needs serious English language and grammar editing.

2. The flow of the manuscript also needs improvement. For example, the authors talk about the aim of their review nearly one third of the way into the manuscript.

3. Many of the figure labels are blurred so are not readable. Therefore, the figure resolution needs to be improved.

4. How are biosensors an improvement over PCR-based techniques for the detection of leukemia molecular changes? The authors state that drawbacks to PCR-based detection are false positives and negatives. However, with appropriate controls, these would be rare. Moreover, as DNA biosensors rely on much the same underlying molecular biology they would also share many of the same drawbacks. Also, basic PCR-based techniques are not complex and do not require instrumentation that would be considered sophisticated by today’s standards. How would sample processing be different for the biosensors discussed, or would it? Are the improvements in sensitivity offered by biosensors enough to detect minimal disease burden? Tables that compare/contrast specifics about the current detection techniques with biosensors could be helpful.

5. Sometimes it is unclear how nanomaterials are improving biosensor capabilities because the authors tend to speak very generally about these nanomaterials. When detail is presented, it can be difficult to follow due to the writing problems. Tables that compare/contrast these nanomaterials and their chemistries could be helpful.

6. The authors refer to APL as the M3 subtype of AML. This is the FAB classification system, which is outdated and based more on morphological features. The WHO classification is the primary system used today and better accounts for molecular abnormalities including gene mutations and chromosomal alterations.

7. The authors state that quantum dots are recognized as biocompatible. This is not entirely true as semiconductor materials are known for their physiological toxicity. However, improvements have been made to develop biocompatible quantum dots, which have biocompatible polymeric shells. Similar toxicity issues exist with many, but not all, nanomaterials. This may not be a big issue for a biosensor using collected bio-specimens but could be if true in vivo use is proposed. However, the mention of biocompatibility in a scientific review could be misinterpreted by general audiences and so the authors should be more specific. It may make sense to include a section on nanotoxicology.

8. TNF is not used in combination with dasatinib for the treatment of CML. It is not the same thing as TRAIL. It is also not used clinically. In contrast, there are clinically-available therapeutics that are used to treat TNF-based inflammatory pathologies.

9. The future perspectives about miRNA detection is confusing due to very writing and unsupported generalizations. The same could be said for most of this section, which is sparse on background information on the numerous topics covered (miRNAs, non-Hodgkin lymphoma, cfDNA).

Author Response

Reviewer 3

The manuscript by Allegra et al. is a review of electrochemical biosensors for leukemia. There is also a focus on nanomaterials being used in these biosensors. Overall, the ideas and concepts are important and worthy of review. However, this manuscript is poorly written and difficult to follow. The following are more specific critiques.

  1. The manuscript is very poorly written, which makes it difficult to understand. This is the case throughout the manuscript, but is worse in the abstract, introduction, and conclusions sections. The entire manuscript needs serious English language and grammar editing.

We subjected the work to linguistic editing

  1. The flow of the manuscript also needs improvement. For example, the authors talk about the aim of their review nearly one third of the way into the manuscript.

We have changed the text and clarified the scope of our work in the first part of the introduction

  1. Many of the figure labels are blurred so are not readable. Therefore, the figure resolution needs to be improved.

 We have modified the figures by increasing their resolution

  1. How are biosensors an improvement over PCR-based techniques for the detection of leukemia molecular changes? The authors state that drawbacks to PCR-based detection are false positives and negatives. However, with appropriate controls, these would be rare. Moreover, as DNA biosensors rely on much the same underlying molecular biology they would also share many of the same drawbacks. Also, basic PCR-based techniques are not complex and do not require instrumentation that would be considered sophisticated by today’s standards. How would sample processing be different for the biosensors discussed, or would it? Are the improvements in sensitivity offered by biosensors enough to detect minimal disease burden? Tables that compare/contrast specifics about the current detection techniques with biosensors could be helpful.

 We have modified the text by reporting other characteristics that could make the use of electrochemical biosensors useful. Furthermore, in the conclusions we reported Li's work on the sensitivity of the various diagnostic techniques in identifying minimal residual disease.

  1. Sometimes it is unclear how nanomaterials are improving biosensor capabilities because the authors tend to speak very generally about these nanomaterials. When detail is presented, it can be difficult to follow due to the writing problems. Tables that compare/contrast these nanomaterials and their chemistries could be helpful.

 The detection limits of the various nanomaterials are often reported in the tables and sometimes in the text. We therefore considered it repetitive to insert a special table. However, we are ready to insert it if the reviewer deems it necessary.

  1. The authors refer to APL as the M3 subtype of AML. This is the FAB classification system, which is outdated and based more on morphological features. The WHO classification is the primary system used today and better accounts for molecular abnormalities including gene mutations and chromosomal alterations.

We have deleted the reference to the FAB classification and referred to the WHO classification.

  1. The authors state that quantum dots are recognized as biocompatible. This is not entirely true as semiconductor materials are known for their physiological toxicity. However, improvements have been made to develop biocompatible quantum dots, which have biocompatible polymeric shells. Similar toxicity issues exist with many, but not all, nanomaterials. This may not be a big issue for a biosensor using collected bio-specimens but could be if true in vivo use is proposed. However, the mention of biocompatibility in a scientific review could be misinterpreted by general audiences and so the authors should be more specific. It may make sense to include a section on nanotoxicology. 

We have deleted the mention of biocompatibility of the quantum dots.

  1. TNF is not used in combination with dasatinib for the treatment of CML. It is not the same thing as TRAIL. It is also not used clinically. In contrast, there are clinically-available therapeutics that are used to treat TNF-based inflammatory pathologies.

We have changed the text and deleted the mention to the TNF

  1. The future perspectives about miRNA detection is confusing due to very writing and unsupported generalizations. The same could be said for most of this section, which is sparse on background information on the numerous topics covered (miRNAs, non-Hodgkin lymphoma, cfDNA).

We deleted some paragraphs trying to make the section less speculative

Round 2

Reviewer 1 Report

 Authors full replied ti criticismo. No other queries

Author Response

Thank you.

Reviewer 2 Report

The authors have addressed all the queries raised. Recommended for Publication.

Author Response

Thank you.

Reviewer 3 Report

The manuscript has been substantially improved. However, there are still some minor English language issues. These are less frequent and mostly in sections 6 and 7.

Author Response

English language issues have been corrected and indicated with blue highligths or strikethrough. Thank you for your contribute.